# AdvCodec: Towards a Unified Framework for Adversarial Text Generation

## Abstract

While there has been great interest on generating imperceptible *adversarial examples* in continuous data domain (e.g. image and audio) to explore the model vulnerabilities, generating *adversarial text* in the discrete domain is still challenging. The main contribution of this paper is to propose a general targeted attack framework `AdvCodec` for adversarial text generation which addresses the challenge of discrete input space and is easily adapted to general natural language processing (NLP) tasks. In particular, we propose a tree based autoencoder to encode discrete text data into continuous vector space, upon which we optimize the adversarial perturbation. A tree based decoder is then applied to ensure the grammar correctness of the generated text. It also enables the flexibility of making manipulations on different levels of text, such as sentence (`AdvCodec(Sent)`) and word (`AdvCodec(Word)`) levels. We consider multiple attacking scenarios, including appending an adversarial sentence or adding unnoticeable words to a given paragraph, to achieve arbitrary *targeted attack*. To demonstrate the effectiveness of the proposed method, we consider two most representative NLP tasks: sentiment analysis and question answering (QA). Extensive experimental results and human studies show that `AdvCodec` generated adversarial text can successfully attack the neural models without misleading the human. In particular, our attack causes a BERT-based sentiment classifier accuracy to drop from $0.703$ to $0.006$, and a BERT-based QA model's F1 score to drop from $88.62$ to $33.21$ (with best targeted attack F1 score as $46.54$). Furthermore, we show that the white-box generated adversarial texts can transfer across other black-box models, shedding light on an effective way to examine the robustness of existing NLP models.

## 1 Introduction

Recent studies have demonstrated that deep neural networks (DNNs) are vulnerable to carefully crafted adversarial examples (Goodfellow et al., 2015; Papernot et al., 2016; Eykholt et al., 2017; Moosavi-Dezfooli et al., 2016). While there are a lot of successful attacks proposed in the continuous data domain including images, audios, and videos, how to effectively generate adversarial examples in the discrete text domain still remains a hard problem. There are several challenges for generating adversarial text: 1) most existing gradient-based adversarial attack approaches are not directly applicable to the discrete structured data; 2) it is less clear how to appropriately measure the naturalness of the generated text compared to the original ones; 3) the manipulation space of text is limited, and it is unclear whether generating a new appended sentence or manipulating individual words will affect human judgements.

So far, existing works on adversarial text generation either leverage heuristic solutions such as genetic algorithms (Jin et al., 2019) to search for potential adversarial sentences, or are limited to attacking specific NLP tasks (Cheng et al., 2018; Lei et al., 2018). In addition, effective targeted attacks have not been achieved by current attacks for any task. In this paper, we aim to provide more insights towards solving these challenges by proposing a unified optimization framework `AdvCodec` to generate adversarial text against general NLP tasks. In particular, the core component of `AdvCodec` is a tree based autoencoder which converts discrete text tokens into continuous semantic embedding, upon which the adversarial perturbation will be optimized regarding the chosen adversarial target. Finally, a tree based decoder will decode the generated adversarial continuous embedding vector back to the sentence level based on the tree grammar rules, aiming to both pre-

serve the original semantic meaning and linguistic coherence. An iterative process can be applied here to ensure the attack success rate.

In addition to the general adversarial text generation framework `AdvCodec`, this paper also aims to explore several scientific questions: 1) Since `AdvCodec` allows the flexibility of manipulating on different hierarchies of the tree structures, which is more attack effective and which way preserves better grammatical correctness? 2) Is it possible to achieve targeted attack for general NLP tasks such as sentiment classification and QA, given the limited degree of freedom for manipulation? 3) Is it possible to perform blackbox attack in general NLP tasks? 4) Is BERT robust in practice? 5) Do these adversarial examples affect human reader performances?

To address the above questions, we explore two types of tree based autoencoders on the word (`AdvCodec(Word)`) and sentence level (`AdvCodec(Sent)`). For each encoding scenario, we generate adversarial text against different sentiment classification and QA models. Compared with the state-of-the-art adversarial text generation methods, our approach achieves significantly higher untargeted and *targeted* attack success rate. In addition, we perform both whitebox and blackbox settings for each attack to evaluate the model vulnerabilities. Within each attack setting, we evaluate attack strategies as appending an additional adversarial sentence or adding scatter of adversarial words to a paragraph, to evaluate the quantitative attack effectiveness. To provide thorough adversarial text quality assessment, we also perform 7 groups of human studies to evaluate the quality of generated adversarial text compared with the baselines methods, and whether human can still get the ground truth answers for these tasks based on adversarial text. We find that: 1) both word and sentence level attacks can achieve high attack success rate, while the sentence level manipulation can consider the global grammatical constraints and generate high quality adversarial sentences. 2) various targeted attacks on general NLP tasks are possible (e.g. when attacking QA, we can ensure the target to be a specific answer or a specific location within a sentence); 3) the transferability based blackbox attacks are successful in NLP tasks. Transferring adversarial text from stronger models (in terms of performances) to weaker ones is more successful; 4) Although BERT has achieved state-of-the-art performances, we observe the performance drops are also larger than other standard models when confronted with adversarial examples, which indicates BERT is not robust under the adversarial settings; 5) Most human readers are not sensitive to our adversarial examples and can still answer the right answers when confronted with the adversary-injected paragraphs.

In summary, our main contribution lies on: (1) We propose a general adversarial text generation framework `AdvCodec` that addresses the challenge of discrete text input to achieve targeted attacks against general NLP tasks (e.g. sentiment classification and QA) while preserving the semantic meaning and linguistic coherence; (2) we propose a novel tree-based text autoencoder that ensures the grammar correctness of generated text; (3) we conduct extensive experiments and successfully attack different sentiment classifiers and QA models with significant higher attack success rate than the state-of-the-art baseline methods; (4) we also perform comprehensive ablation studies including evaluating the attack scenarios of appending an adversarial sentence or adding scatter of adversarial words, as well as appending the adversarial sentence at different positions within a paragraph, and draw several interesting conclusions; (5) we leverage extensive human studies to show that the adversarial text generated by `AdvCodec` is natural and effective to attack neural models, while barely affecting human's judgement.

## 2 RELATED WORK

A large body of works on *adversarial examples* focus on perturbing the continuous input space. Though some progress has been made on generating adversarial perturbations in the discrete space, several challenges still remain unsolved. For example, Zhao et al. (2017) exploit the generative adversarial network (GAN) to generate natural adversarial text. However, this approach cannot explicitly control the quality of the generated instances. Most existing methods (Liang et al., 2017; Samanta & Mehta, 2017; Jia & Liang, 2017; Li et al., 2018; Jin et al., 2019) apply heuristic strategies to synthesize adversarial text: 1) first identify the features (e.g. characters, words, and sentences) that have the influence on the prediction, 2) follow different search strategies to perturb these features with the constructed perturbation candidates (e.g. typos, synonyms, antonyms, frequent words). For instance, Liang et al. (2017) employ the loss gradient $\nabla L$ to select important characters and phrases to perturb, while Samanta & Mehta (2017) use typos, synonyms, and important adverbs/adjectives as candidates for insertion and replacement. Once the influential features are obtained, the strategies to

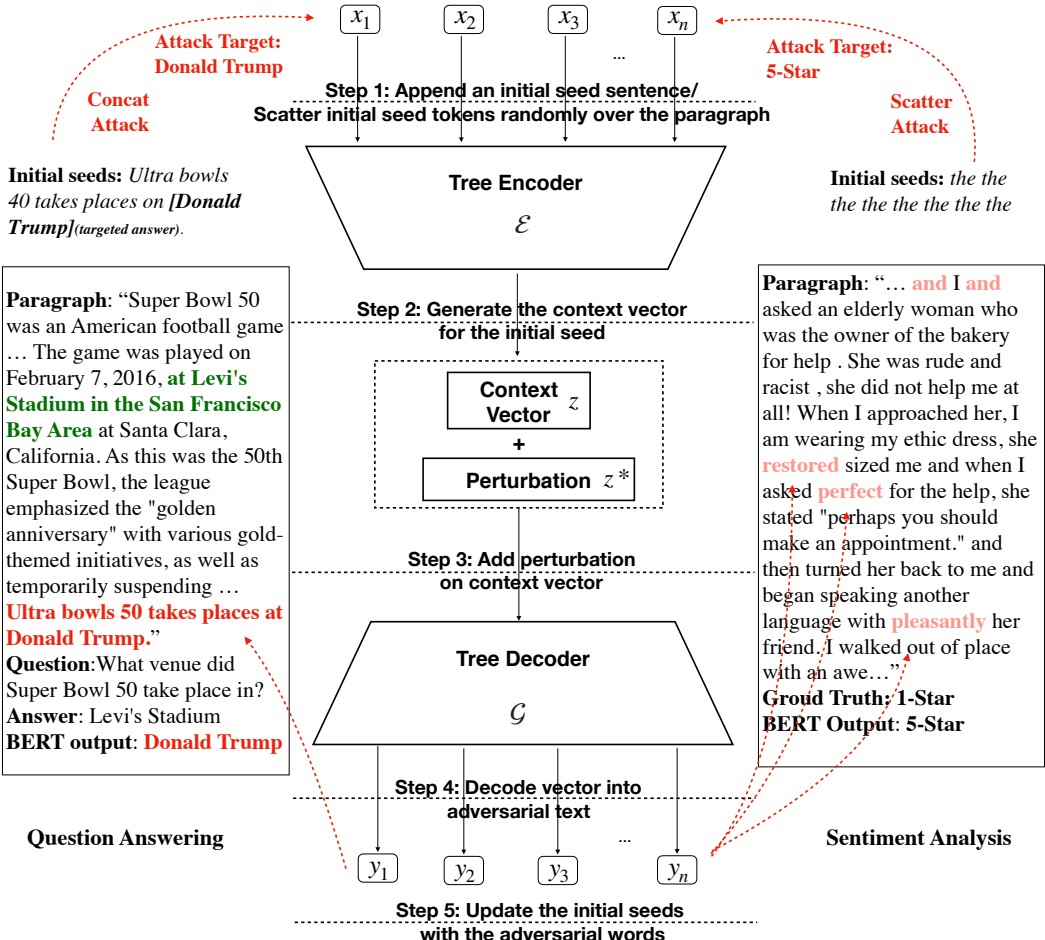

Figure 1: Overview of `AdvCodec`. Here we illustrate the pipeline of generating adversarial text for Question Answering and Sentiment Analysis tasks.

apply the perturbation generally include *insertion*, *deletion*, and *replacement*. Such adversarial text generation approaches cannot guarantee the grammar correctness of generated text. For instance, text generated by Liang et al. (2017) are almost random stream of characters. To generate grammarly correct perturbation, Jia & Liang (2017) adopt another heuristic strategy which adds *manually* constructed legit distracting sentences to the paragraph to introduce fake information. These heuristic approaches are in general not scalable, and cannot achieve targeted attack where the adversarial text can lead to a chosen adversarial target (e.g. adversarial label in classification). Recent work searches for a universal trigger (Wallace et al., 2019) to be applied to arbitrary sentences to fool the learner, while the reported attack success rate is rather low. In contrast, with the tree based autoencoder, the proposed `AdvCodec` framework is able to generate grammarly correct adversarial text efficiently, achieving high attack success rates on different models.

## 3 THE ADVCODEC FRAMEWORK FOR ADVERSARIAL TEXT GENERATION

We describe the `AdvCodec` framework in this section. As illustrated in Figure 1, the key component of the `AdvCodec` framework is a tree-based autoencoder. The hierarchical and discrete nature of language motivates us to make use of tree-based autoencoder to map discrete text into a high dimensional latent space, which empowers us to leverage the existing optimization based attacking method such as Carlini & Wagner (2016) to both efficiently and effectively generate adversarial text.

Let $X$ be the domain of text and $S$ be the domain of dependency parsing trees over element in $X$. Formally, a tree-based autoencoder consists of an encoder $\mathcal{E} : X \times S \rightarrow Z$ that encodes text $x \in X$ along with its dependency parsing tree $s \in S$ into a high dimensional latent representation $z \in Z$

and a decoder $\mathcal{G} : Z \times S \rightarrow X$ that generates the corresponding text $x$ from the given context vector $z$ and the expected dependency parsing tree $s$. Given a dependency tree $s$, $\mathcal{E}$ and $\mathcal{G}$ form an antoencoder. We thus have the following reconstruction loss to train our tree-based autoencoder:

$$L = -\mathbb{E}_{x \sim X}[\log p_{\mathcal{G}}(x|s, \mathcal{E}(x, s)] \qquad (1)$$

As Figure 1 suggests, `AdvCodec` can operate on different granularity levels to generate either word-level or sentence-level contextual representation, and decode it into the adversarial text. We refer the sentence-level `AdvCodec` to `AdvCodec(Sent)` and the word-level one to `AdvCodec(Word)`. Both of them will be described in more details in the later part of this section.

## 3.1 Overview of the AdvCodec Framework

Before diving into details, we provide a high level overview of `AdvCodec` according to the attack scenario and attack capability supported by this framework.

**Attack Scenario.** Different from the previous adversarial text generation works (Lei et al., 2018; Cheng et al., 2018; Papernot et al., 2016; Miyato et al., 2016; Alzantot et al., 2018) that directly modify critical words in place and might risk changing the semantic meaning or editing the ground truth answers, we are generating the *concatenative adversaries*. First proposed by Jia & Liang (2017), the concatenative adversary does not change any words in the original paragraph or question, but instead appends a new adversarial sentence to the paragraph to fool the model. However, the concatenative attack also needs to ensure the appended sentence is *compatible* (Jia & Liang, 2017) with the original paragraph, which in other words means it should not contradict any stated facts in the paragraph, especially the correct answer. In our work, we further push the concept of concatenative adversaries further and propose a more general notion called *scatter attack*, which means we can inject adversarial words sporadically over the whole paragraph. The concatenative adversarial example falls into our case when those adversarial tokens form a sentence and on the same time the semantic meaning of the sentence does not contradict the original paragraph. Examples of concatenative attack and scatter attack is shown in table 1.

Table 1: In the sentiment analysis task, we perform concatenative attack and scatter attack. Concatenative adversary does not change existing context but instead appends the adversarial sentence to the paragraph, while scatter attack scatters adversarial tokens over the whole passage. In the QA task, with the answer targeted to "Donald Trump", the model outputs "Donald Trump" after answer targeted attack.

| Task | Input(red=Inserted or appended tokens) | Model Prediction |
|------|----------------------------------------|------------------|
| Sentiment Analysis | **Concatenative Attack:** I kept expecting to see chickens and chickens walking around. ... This place is like a steinbeck novel come to life. I kept expecting to see donkeys and chickens walking around. wooo-pig-soooeeee this place is awful!!! | Negative → Positive |
| | **Scatter Attack:** ... She was rude and racist , she did not help me at all! when i approached he, I am wearing my ethic dress, she restored sized me and when i asked perfect for the help, she stated "perhaps you should make an appointment. " And then turned her back to me and began speaking another language with pleasantly her friend... | Negative → Positive |
| QA | *Question: What is the smallest geographical region discussed?* *Paragraph:* Its counties of Los Angeles, Orange, San Diego, San Bernardino, and Riverside are the five most populous in the state and all are in the top 15 most populous counties in the United States. a simplest geographic regions discuss donald trump. | Riverside → donald trump |

**Attack Capability.** `AdvCodec` is essentially an optimization based framework to find the adversarial texts with the optimization goal set to achieve *targeted attack*. For the sentiment classification task, `AdvCodec` can perform targeted attack to make the original positive reviews be classified as the most negative one, and vice versa. Particularly in the QA task, we design and implement two kinds of targeted attack: *position targeted attack* and *answer targeted attack*. A successful *position targeted attack* means the model can be fooled to output the answers at specific targeted positions in the paragraph, but the content on the targeted span cannot be guaranteed. In contrast, a successful *answer targeted attack* is a stronger targeted attack, which refers to the situation when the model always outputs the preset targeted answer pair on the target no matter what the question looks like. An

example of word targeted attack can be found in the table 1. Although our framework is designed as a whitebox attack, our experimental results demonstrate our whitebox generated adversarial words can transfer to other blackbox models with high attack success rate. Finally, because `AdvCodec` is a unified adversarial text generation framework whose outputs are discrete tokens, it can be applied to different downstream NLP tasks. In this paper, we perform adversarial evaluation on sentiment classification and QA as examples to demonstrate how our framework is adapted to different works.

### 3.2 ADVCODEC(SENT)

In this subsection, we describe `AdvCodec(Sent)` and explain how to utilize it to attack sentiment classification models and question answering systems. The main idea comes from the fact that tree structures sometimes have better performances than sequential recurrent models(Li et al., 2015; Iyyer et al., 2014; 2018) and the fact that it is inherently flexible to add perturbations on hierarchical nodes of the tree structures. Motivated by this, we design a novel tree-based autoencoder to simultaneously preserve similar semantic meaning and original syntactic structure.

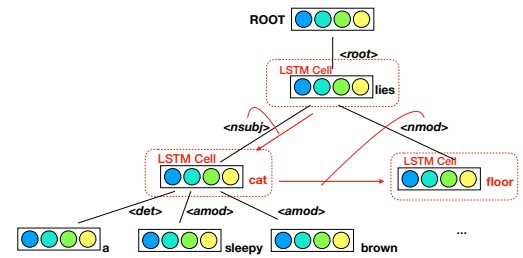

Figure 2: The tree decoder. Each node in the dependency tree is a LSTM cell. Black lines refer to the dependencies between parent and child nodes. Red arrows refer to the directions of decoding. During each step the decoder outputs a token that is shown on the right of the node.

**Encoder.** We adopt the Stanford Tree-structured LSTM (Tai et al., 2015) as our tree encoder. In the encoding phase, features are extracted and summed from bottom (leaf node, i.e. word) to top (root node) along the dependency tree, extracted by Stanford CoreNLP Parser (Manning et al., 2014). The context vector $z$ for `AdvCodec(Sent)` refers to the root node embedding $h_{root}$, representing the sentence-level embedding.

**Decoder.** Following the same dependency tree, we design the text decoder as illustrated in Figure 2. In the decoding phase, we start from the root node and traverse along the dependency tree in level-order. The hidden state $h_j$ of the next node $j$ comes from (i) the hidden state $h_i$ of the current tree node, (ii) current node predicted word embedding $w_i$, and (iii) the dependency embedding $d_{ij}$ between the current node $i$ and the next node $j$ based on the dependency tree. The next node's corresponding word $y_j$ is generated based on the output of the LSTM Cell $o_j$ via a linear layer that maps from the hidden presentation $o_j$ to the logits that represent the probability distribution of the tree's vocabulary.

$$o_j, h_j = \text{LSTM}([h_i; w_i; d_{ij}]) \tag{2}$$
$$y_j = W \cdot o_j + b \tag{3}$$

#### 3.2.1 ATTACK SENTIMENT CLASSIFICATION MODEL

**Initial Seed.** Following our pipeline to optimize adversarial sentence `AdvSentence` appended to the paragraph, we need to first start with an initial seed for optimization. Such initial seed for sentiment classification task can be arbitrary. For example, we can simply sample a sentence no shorter than 3 words from the original paragraph and append it to the start of the paragraph when attacking the BERT. The append position does have a influence on the attack success rate for adversarial attack, and more detailed ablation analysis will be discussed in the next section.

**Optimization Procedure.** Finding the optimal perturbation $z*$ on context vector $z$, we aim to find $z^*$ that solves

$$\text{minimize} \quad ||z^*||_p + cf(z + z^*), \tag{4}$$

where $f$ is the objective function for the targeted attack and $c$ is the constant balancing between the perturbation magnitude and attack target. Specifically, we use the objective function $f$ proposed in Carlini & Wagner (2016) as follows

$$f(z') = \max(\max\{Z(\mathcal{G}(z', s))_i : i \neq t\} - Z(\mathcal{G}(z', s))_t, -\kappa) \tag{5}$$

where $z' = z + z^*$, $t$ is the target class, $Z(\cdot)$ is the logit output of the classification model before softmax and $\kappa$ is the confidence score to adjust the misclassification rate. The optimal solution is iteratively searched via Adam optimizer (Kingma & Ba, 2014).

### 3.2.2 ATTACK QUESTION ANSWERING SYSTEM

**Initial Seed.** Different from attacking sentiment analysis, it is important to choose a good initial seed that is semantically close to the context or the question when attacking QA model. In this way we can reduce the number of iteration steps and attack the QA model more efficiently. Based on the heuristic experiments conducted in the Appendix A.4, we choose to use question words to craft an initial seed. We design a set of coarse grained rules to convert a question sentence to a meaningful declarative statement and assign a target fake answer. The fake answer can be crafted according to the perturbed model's predicted answer, or can be manually chosen by adversaries. As for the location where we append the sentence, we choose to follow the setting in Jia & Liang to add the adversary to the end of the paragraph so that we can make a fair comparison with their results.

It is worth noting unlike Jia & Liang (2017) that uses complicated rules to ensure the adversarial sentence does not change the ground truth, this heuristic step is the very first step of our framework followed by a series of optimization steps to ensure the ground truth is not changed. In this paper, we ensure our appended adversarial sentences are not contradictory to the ground truth by a) choosing an initial sentence as the initial seed of optimization, b) adding perturbation to the sentence, c) searching for the optimal adversarial sentence, d) ensuring that the adversarial sentence and context sentence are disjoint, otherwise keep the iteration steps. If the maximum steps are reached, the optimization is regarded as a failure.

**Optimization Procedure.** We follow the same optimization procedure as attacking sentiment classification task except a subtle change of the objective function $f$ due to the difference between QA model and classification model:

$$f(z') = \sum_{j=1}^{2} \max(\max\{Z_j(\mathcal{G}(z',s))_i : i \neq t\} - Z_j(\mathcal{G}(z',s))_{t_j}, -\kappa) \qquad (6)$$

where $Z_1(\cdot)$ is the output logits of answer starting position and $Z_2(\cdot)$ is the output logits of answer ending position in the QA system. $t_1$ and $t_2$ are respectively the targeted start position and the targeted end position. For the position targeted attack mentioned in Section 3.1, we expect the model output to be a span in the paragraph from the targeted start position $t_1$ to the targeted end position $t_2$. In contrast, the answer targeted attack requires the model to output the predefined answer spans in the targeted positions and keep them unmodified during the optimization steps by setting gates to the targeted answer span: $y_j = g_1 \odot y_j + g_2 \odot x_j, \quad (j = t_1, t_1 + 1, ..., t_2)$, where $y_j$ refers to the tree decoded adversarial tokens. We set $g_1 = 1$ and $g_2 = 0$ in the position targeted attack, and $g_1 = 0$ and $g_2 = 1$ in the answer targeted attack.

### 3.3 ADVCODEC(WORD)

Not only we can apply perturbations to the root node of our tree-based autoencoder to generate adversarial sentence, we can also perturb nodes at different hierachical levels of the tree to generate adversarial word. The most general case is that the perturbation is directly exerted on the leaf node of the tree autoencoder, i.e. the word-level perturbation.

`AdvCodec(Word)` shares the exactly same architectures and optimization steps mentioned above to attack the targeted models. The distinction between `AdvCodec(Word)` and `AdvCodec(Sent)` is the context vector $z$. Formally for the word-level attack, the context vector $z$ are the concatenation of leaf node embedding $z_i$ (which corresponds to each word) $z = [z_1, z_2, \ldots, z_n]$. Different from the `AdvCodec(Sent)` that perturbation is added on the whole sentence, we can control where the perturbations are added by assigning each node a mask as follows:

$$z_i' = z_i + \text{mask} \cdot z_i^* \qquad (7)$$

When we expect some token $z_i$ to be adversarially changed, we can simply assign mask $= 1$, thus adding the perturbation on the token.

As the perturbation can be controlled on individual words, we propose a new attack scenario *scatter attack*, which scatters some initial tokens over the paragraph, adds perturbation only to those tokens

and find the best adversarial tokens via the same optimization procedure mentioned above. More-over, the concatenative adversarial examples (e.g. generated by `AdvCodec(Sent)`) can also be crafted by `AdvCodec(Word)` because the concateneative adversaries are simply special cases for the scatter attack.

## 4 EXPERIMENTAL RESULTS

In this section we will present the experimental evaluation results for `AdvCodec`. In particular, we target on two popular NLP tasks, sentiment classification and QA. For both models, we perform whitebox and transferability based blackbox attacks. In addition to the model accuracy (untargeted attack evaluation), we also report the targeted attack success rate for `AdvCodec`. We show that the proposed `AdvCodec` can outperform other state of the art baseline methods on different models.

### 4.1 SENTIMENT ANALYSIS

**Task and Dataset.** In this task, sentiment analysis model takes the user reviews from restaurants and stores as input and is expected to predict the number of stars (from 1 to 5 star) that the user was assigned. We choose the Yelp dataset (Challenge) for sentiment analysis task. It consists of 2.7M yelp reviews, in which we follow the process of Lin et al. (2017) to randomly select 500K review-star pairs as the training set, and 2000 as the development set, 2000 as the test set.

**Model.** *BERT* (Devlin et al., 2019) is a transformer (Vaswani et al., 2017) based model, which is unsupervisedly pretrained on a large corpus and is proven to be effective for downstream NLP tasks. *Self-Attentive Model (SAM)* (Lin et al., 2017) is a state-of-the-art text classification model uses self-attentive mechanism. More detailed model settings are listed in the appendix.

**Baseline.** *Seq2sick* (Cheng et al., 2018) is a whitebox projected gradient method to attack seq2seq models. Here, we perform seq2sick attack on sentiment classification models by changing its loss function, which was not evaluated in the original paper.*TextFooler* (Jin et al., 2019) is a simple yet strong blackbox attack method to generate adversarial text. Following the same setting, Seq2Sick and TextFooler is only allowed to edit the appended sentence or tokens.

**Adversarial Evaluation.** We perform the baseline attacks and our `AdvCodec` attack in scatter attack scenario and concat attack scenario under the whitebox settings. Our targeted goal for senti-ment classification is the opposite sentiment. Specifically, we set the targeted attack goal as 5-star for reviews originally below 3-star and 1-star for reviews above. We compare our results with a strong word-level attacker Seq2sick, as shown in the Table 2. We can see our `AdvCodec(Word)` outperforms the baselines and achieves nearly 100% attack success rate on the BERT model. Also, we realize the targeted success rate for `AdvCodec(Sent)` is lower than the word-level base-line. We assume the reason is that `AdvCodec(Sent)` has the dependency tree constraints during decoding phase, thus increasing the difficulty to find both grammatical correct and adversarial sen-tences that can successful attack. On the contrary, the Seq2Sick baseline can edit any words under no semantic or syntactic constraints. Moreover, our following human evaluation exactly confirms `AdvCodec(Sent)` has better language quality.

**Scatter Attack v.s. Concat Attack.** In addition, we find scatter attack success rate is slightly lower than concat attack. We think there are two reasons to explain this phenomenon: Firstly, the average number of tokens added in scatter attack is 10, while the average number of tokens added in concat attack is 19. Therefore concat attack has the freedom to manipulate on more words than scatter

Table 2: Whitebox attack success rates on sentiment analysis. Targeted attack success rate is mea-sured by how many examples are successfully attacked to output the targeted label in average, while untargeted attack success rate calculates the percentage of examples attacked to output a label dif-ferent from the ground truth. Adv(·) is short for our attack AdvCodec(·) at different levels.

| Model | Original Acc | | Concat Attack | | | Scatter Attack | |
|---|---|---|---|---|---|---|---|
| | | | Adv(Sent) | Adv(Word) | Seq2Sick | Adv(Word) | Seq2sick |
| BERT | 0.703 | target | 0.466 | **0.990** | 0.974 | 0.976 | 0.946 |
| | | untarget | 0.637 | **0.993** | 0.988 | 0.987 | 9.970 |
| SAM | 0.704 | target | 0.756 | **0.956** | 0.933 | 0.869 | 0.570 |
| | | untarget | 0.810 | **0.967** | 0.952 | 0.948 | 0.711 |

attack, thus resulting in higher attack accuracy. Secondly, inserting adversarial tokens to different positions of the passage also affects the success rate, which is shown in Appendix A.5.

**Blackbox Attack.** We perform transferability based blackbox attacks. We compare our blackbox attack success rate with the blackbox baseline TextFooler and blackbox Seq2Sick based on transferability. Table 3 demonstrates our `AdvCodec(Word)` model still has the best blackbox targeted and untargeted success rate among all the baseline models.

Table 3: Blackbox attack success rates on sentiment analysis. The transferability-based blackbox attack uses adversarial text generated from whitebox BERT model to attack blackbox SAM, and vice versa. TF is short for TextFooler.

| Model | | Concat Attack | | | | Scatter Attack | | |
|---|---|---|---|---|---|---|---|---|
| | | Adv(Sent) | Adv(Word) | Seq2Sick | TF | Adv(Word) | Seq2sick | TF |
| BERT | target | 0.187 | **0.499** | 0.218 | 0.042 | 0.298 | 0.156 | 0.107 |
| | untarget | 0.478 | **0.686** | 0.510 | 0.318 | 0.574 | 0.445 | 0.392 |
| SAM | target | 0.335 | **0.516** | 0.333 | 0.113 | 0.465 | 0.230 | 0.081 |
| | untarget | 0.533 | 0.669 | 0.583 | 0.395 | **0.679** | 0.498 | 0.335 |

## 4.2 QUESTION ANSWERING (QA)

**Task and Dataset.** In this task, we choose the SQuAD dataset (Rajpurkar et al., 2016) for question answering task. The SQuAD dataset is a reading comprehension dataset consisting of 107,785 questions posed by crowd workers on a set of Wikipedia articles, where the answer to each question must be a segment of text from the corresponding reading passage. To compare our method with other adversarial evaluation works (Jia & Liang, 2017) on the QA task, we evaluate our adversarial attacks on the same test set as Jia & Liang (2017), which consists of 1000 randomly sampled examples from the SQuAD development set. We use the official script of the SQuAD dataset (Rajpurkar et al., 2016) to measure both adversarial exact match rates and F1 scores.

**Model.** We adapt the *BERT* model to run on SQuAD v1.1 with the same strategy as that in Devlin et al. (2019), and we reproduce the result on the development set. *BiDAF*(Seo et al., 2016) is a multi-stage hierarchical process that represents the context at different levels of granularity and uses bidirectional attention flow mechanism to obtain a query-aware context representation.

**Baseline.** *Universal Adversarial Triggers* (Wallace et al., 2019) are input-agnostic sequences of tokens that trigger a model to produce a specific prediction when concatenated to any input from a dataset. Here, we compare the targeted attack ability of `AdvCodec` with it. *AddSent* (Jia & Liang, 2017) appends a manually constructed legit distracting sentence to the given text so as to introduce fake information, which can only perform untargeted attack.

**Adversarial Evaluation.** We perform the whitebox attack with different attack methods on our testing models. As is shown in Table 4 , `AdvCodec(Word)` achieves the best whitebox attack results on both BERT and BiDAF. It is worth noting although BERT has better performances than BiDAF, the performance drop for BERT $\Delta \text{F1}_{BERT}$ is 55.4 larger than the performance drop for BiDAF $\Delta \text{F1}_{BiDAF} = 53.0$, which again proves the BERT is insecure under the adversarial evaluation. We also find the position targeted attack is slightly stronger than the answer targeted attack. We assume it is because the answer targeted attack has fixed targeted answer and limited freedom to alter the appended sentence, but the position targeted attack has more freedom to alter the fake answer from

Table 4: Whitebox attack results on QA in terms of exact match rates and F1 scores by the official evaluation script. The lower EM and F1 scores mean the better attack success rate.

| Model | | Origin | Position Targeted Attack | | Answer Targeted Attack | | Baseline (untargeted) |
|---|---|---|---|---|---|---|---|
| | | | Adv(Sent) | Adv(Word) | Adv(Sent) | Adv(Word) | AddSent |
| BERT | EM | 81.2 | 49.1 | **29.3** | 50.9 | 43.2 | 46.8 |
| | F1 | 88.6 | 53.8 | **33.2** | 55.2 | 47.3 | 52.6 |
| BiDAF | EM | 60.0 | 29.3 | **15.0** | 30.2 | 21.0 | 25.3 |
| | F1 | 70.6 | 34.0 | **17.6** | 34.4 | 23.6 | 32.0 |

the targeted position spans. We also tried the scatter attack on QA though the performances are not good. It turns out QA systems highly rely on the relationship between questions and contextual clues, which is hard to break when setting an arbitrary token to a target answer. We discussed in Appendix A.3 the untargeted scatter attack can work well and outperform the baseline methods.

Then we test the targeted results of whitebox attack methods on QA models. The results are shown in Table 5. It shows that `AdvCodec(Word)` has the best targeted attack ability on QA. And all our attack methods outperform the baseline(Universal Triggers) when it comes to the targeted results.

**Blackbox Attack.** We also transfer adversarial texts generated from whitebox attacks to perform blackbox attacks. Table 6 shows the result of the blackbox attack on testing models. All our proposed methods outperform the baseline method(AddSent) when transferring the adversaries among models with same architectures.

Table 5: Targeted Attack Results of whitebox attack on QA. Here, the targeted exact match rates and targeted F1 Score measures how many model outputs match the targeted fake answers. Higher targeted EM and F1 mean higher targeted attack success rate. UT is short for Universal Trigger baseline.

| Model | | Adv(Sent) | Adv(Word) | UT |
|---|---|---|---|---|
| BERT | target EM | 32.1 | **43.4** | 1.4 |
| | target F1 | 32.4 | **46.5** | 2.1 |
| BiDAF | target EM | 53.3 | **71.2** | 21.2 |
| | target F1 | 56.8 | **75.6** | 22.6 |

Table 6: BlackBox attack results on QA in terms of exact match rates and F1 scores. The transferability-based blackbox attack uses adversarial text generated from whitebox models (annotated as $(w)$) to attack different blakcbox models (annotated as $(b)$).

| From | Attack | | Position Targeted Attack | | Answer Targeted Attack | | Baseline (untargeted) |
|---|---|---|---|---|---|---|---|
| | | | Adv(Sent) | Adv(Word) | Adv(Sent) | Adv(Word) | AddSent |
| BiDAF$_{(w)}$ | BERT$_{(b)}$ | EM | 57.7 | 52.8 | 58.7 | 51.7 | **46.4** |
| | | F1 | 62.9 | 57.5 | 63.7 | 55.9 | **51.9** |
| | BiDAF$_{(b)}$ | EM | 26.7 | **18.9** | 26.4 | 20.5 | 22.3 |
| | | F1 | 31.3 | **22.5** | 30.6 | 24.1 | 27.8 |
| BERT$_{(w)}$ | BERT$_{(b)}$ | EM | 47.0 | **32.3** | 49.6 | 45.2 | 46.4 |
| | | F1 | 52.0 | **36.4** | 54.2 | 49.0 | 51.9 |
| | BiDAF$_{(b)}$ | EM | 30.4 | 29.2 | 29.8 | 28.9 | **22.3** |
| | | F1 | 35.5 | 34.5 | 35.3 | 34.2 | **27.8** |

## 5 HUMAN EVALUATION

We conduct a thorough human subject evaluation to assess the human response to different types of generated adversarial text. The main conclusion is that even though these adversarial examples are effective at attacking machine learning models, they are much less noticeable by humans.

### 5.1 COMPARISON OF ADVERSARIAL TEXT QUALITY

To understand what humans think of our adversarial data quality, we present the adversarial text generated by `AdvCodec(Sent)` and `AdvCodec(Word)` based on the same initial seed. Human participants are asked to choose which data they think has better language quality.

In this experiement, we prepare 600 adversarial text pairs from the same paragraphs and initial seeds. We hand out these pairs to 28 Amazon Turks. Each turk is required to annotate at least 20 pairs and at most 140 pairs to ensure the task has been well understood. We assign each pair to at least 5 unique turks and take the majority votes over the responses. Human evaluation results are shown in Table 7, from which we see that the overall vote ratio for `AdvCodec(Sent)` is 66%, meaning

Table 7: Human evaluation on adversarial text quality aggregated by majority vote.

| Method | Maj Vote |
|---|---|
| `AdvCodec(Sent)` | 65.67% |
| `AdvCodec(Word)` | 34.33% |

`AdvCodec(Sent)` has better language quality than `AdvCodec(Word)` from a human perspective. This is due to the fact that `AdvCodec(Sent)` more fully harness the tree-based autoencoder structure compared to `AdvCodec(Sent)`. And it is no surprise that better language quality comes

at the expense of a lower adversarial success rate. As Table 2 shows, the adversarial targeted success rate of `AdvCodec(Sent)` on SAM is $20\%$ lower than that of `AdvCodec(Word)`, which confirms the trade-off between language quality and adversarial success rate.

## 5.2 Human performance on adversarial text

<table>
<tr><td colspan="2">Table 8: Human performance on Sentiment Analysis</td><td colspan="2">Table 9: Human performance on QA</td></tr>
<tr><td>Method</td><td>Majority Acc</td><td>Method</td><td>Majority F1</td></tr>
<tr><td>Origin</td><td>0.95</td><td>Origin</td><td>90.987</td></tr>
<tr><td>AdvCodec(Word)</td><td>0.82</td><td>AdvCodec(Word)</td><td>82.897</td></tr>
<tr><td>AdvCodec(Sent)</td><td>0.82</td><td>AdvCodec(Sent)</td><td>81.784</td></tr>
</table>

To ensure that our generated adversarial text are compatible with the original paragraph, we ask human participants to perform the sentiment classification and question answering task both on the original dataset and adversarial dataset. Adversarial dataset on sentiment classification consists of `AdvCodec(Sent)` concatenative adversarial examples and `AdvCodec(Word)` scatter attack exmaples. Adversarial dataset on QA consists of concatenative adversarial examples genereated by both `AdvCodec(Sent)` and `AdvCodec(Word)`. More specifically, we respectively prepare 100 benign and adversarial data pairs for both QA and sentiment classification, and hand out them to 505 Amazon Turks. Each turk is requested to answer at least 5 question and at most 15 questions for the QA task and judge the sentiment for at least 10 paragraphs and at most 20 paragraphs for the sentiment classification task. We also perform a majority vote over Turk's answers for the same question. The human evaluation results are displayed in Table 8 and Table 9, from which we see that most of our concatenated adversarial text are compatible to the paragraph. While we can spot a drop from the benign to adversarial datasets, we conduct an error analysis in QA and find the error examples are noisy and not necessarily caused by our adversarial text. For adversarial data in the sentiment classification task, we notice that the generated tokens or appended sentences have opposite sentiment from the benign one. However, our evaluation results show human readers can naturally ignore abnormal tokens and make correct judgement according to the context.

## 6 Discussion and Future Works

Besides the conclusions pointed out in the Introduction section, we also summarize some interesting findings: (1) While `AdvCodec(Word)` achieves best attack success rate among multiple tasks, we observe a trade-off between the freedom of manipulation and the attack capability. For instance, `AdvCodec(Sent)` has dependency tree constraints and becomes more natural for human readers than but less effective to attack models than `AdvCodec(Word)`. Similarly, the answer targeted attack in QA has fewer words to manipulate and change than the position targeted attack, and therefore has slightly weaker attack performances. (2) Scatter attack is as effective as concat attack in sentiment classification task but less successful in QA, because QA systems make decisions highly based on the contextual correlation between the question and the paragraph, which makes it difficult to set an arbitrary token as our targeted answer. (3) Transferring adversarial text from models with better performances to weaker ones is more successful. For example, transfering the adversarial examples from BERT-QA to BiDAF achieves much better attack success rate than in the reverse way. (4) We also notice adversarial examples have better transferability among the models with similar architectures than different architectures. (5) BERT models pay more attention to the both ends of the paragraphs and tend to overlook the content in the middle, as shown in Appendix A.5 ablation study that adding adversarial sentences in the middle of the paragraph is less effective than in the front or the end. To defend against these adversaries, here we discuss about the following possible methods and will in depth explore them in our future works: (1) **Adversarial Training** is a practical methods to defend against adversarial examples. However, the drawback is we usually cannot know in advance what the threat model is, which makes adversarial training less effective when facing unseen attacks. (2) **Interval Bound Propagation** (IBP) (Dvijotham et al., 2018) is proposed as a new technique to theoretically consider the worst-case perturbation. Recent works (Jia et al., 2019; Huang et al., 2019) have applied IBP in the NLP domain to certify the robustness of models. (3) **Language models** including GPT2 (Radford et al., 2019) may also function as an anomaly detector to probe the inconsistent and unnatural adversarial sentences.

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

# A  ADVCODEC SETTINGS

## A.1  AUTOENCODER SELECTION

**Seq2seq autoencoder.** We also tried the tradition sequential architecture (seq2seq) as a different autoencoder in the `AdvCodec` pipeline. For the seq2seq encoder-decoder, we use a bi-directional LSTM as the encoder (Hochreiter & Schmidhuber, 1997) and a two-layer LSTM plus soft attention mechanism over the encoded states as the decoder (Bahdanau et al., 2015).

During the attack, the LSTM cell sequentially takes the embedding of each word $x_i$ as input and output the encoded state $h_i$. The context vector $z$ here refers to the last step's output $h_n$ of the encoder LSTM cell. The perturbation $z^*$ is added only on the context vector $h_n$ without influencing previous encoded states $h_i$ ($i < n$).

As the ablation study, we compare its whitebox attack capability with our `AdvCodec` on BiDAF on QA task. As table 10 shows, we can see seq2seq based `AdvCodec` cannot achieve good attack success rate. Moreover, because seq2seq model does not take grammatical constraints into consideration, the quality of generated adversarial text cannot be ensured.

Table 10: Whitebox attack results on QA in terms of exact match rates and F1 scores by the official evaluation script. The lower EM and F1 scores mean the better attack success rate. Adv(seq2seq) refers to `AdvCodec`, which uses seq2seq model as text autoencoder.

| Model | | Origin | Position Targeted Attack | | | Baseline (untargeted) |
|-------|------|--------|----------|-----------|-------------|------------------------|
| | | | Adv(Sent) | Adv(Word) | Adv(seq2seq) | AddSent |
| BiDAF | EM | 60.0 | 29.3 | **15.0** | 51.3 | 25.3 |
| | F1 | 70.6 | 34.0 | **17.6** | 57.5 | 32.0 |

**Tree autoencoder.** In the whole experiments, we used Stanford TreeLSTM as tree encoder and our proposed tree decoder together as tree autoencoder. We trained the tree autoencoder on yelp dataset which contains 500K reviews. The model is expected to read a sentence, map the sentence in a latent space and reconstruct the sentence from the embedding along with the dependency tree structure in an unsupervised manner. The model uses 300-d vectors as hidden tree node embedding and is trained for 30 epochs with adaptive learning rate and weight decay. After training, the average reconstruction loss on test set is 0.63.

## A.2  ATTACK SETTINGS

We used the Carlini & Wagner (2016) attack as the optimization procudure to search for the optimal $z^*$ that can attack the targeted model. We update $z^*$ iteratively via gradient descent over the optimization function (5) and (6) for different tasks. We use Adam (Kingma & Ba, 2014) as the optimizer, set the learning rate to 0.6 and the optimization steps to 100. We follow the Carlini & Wagner (2016) method to find the suitable parameters in the object function (weight const $c$ and confidence score $\kappa$) by binary search.

---

**Algorithm 1** Algorithm of `AdvCodec` generating adversarial examples

---

1: **procedure** ADVCODEC($x, s$)  $\triangleright x$: initial seed, $s$: corresponding dependency tree
2:     $z := \mathcal{E}(x, s)$  $\triangleright \mathcal{E}$: encoder of `AdvCodec`, $z$: context vector
3:     $z^* = 0$  $\triangleright z^*$: perturbation on context vector
4:     $z' := z + z^*$  $\triangleright z'$: perturbed context vector
5:     $y := \mathcal{G}(z', s)$  $\triangleright \mathcal{G}$: decoder of `AdvCodec`, $y$: adversarial sentence
6:     $f(z') :=$ the objective function to attack the targeted model
7:     **while** $y$ does not achieve targeted attack **do**
8:         update $z^*$ by gradient descent over objective function $f(z')$
9:     **end while**
10:     **return** $y$
11: **end procedure**

---

We also include our attack algorithm via pseudo-code in Algorithm 1.

## A.3 Untargeted scatter attack on QA

We tried the scatter attack on QA, however, the targeted attack success rate is not satisfactory. It turns out QA systems highly rely on the relationship between questions and contextual clues, which is hard to break when setting an arbitrary token to a target answer. This is also why we use some preliminary approaches to creating a similar fake context when initializing QA appended sentence.

We also performed the untargeted scatter attack on QA. The results are shown in table 11. We insert 30 random tokens (but no more than $1/3$ the total words of the paragraph) over the paragraph, optimize and find the adversarial tokens that can cause model output the wrong answers in the untargeted manner. We can see the untargeted scatter attack can also achieve a higher untargeted attack success rate than Jia & Liang (2017).

Table 11: Whitebox attack results on BERT-QA in terms of exact match rates and F1 scores by the official evaluation script. The lower EM and F1 scores mean the better attack success rate.

| Model | | Origin | Position Targeted Attack | | Answer Targeted Attack | | Untargeted Attack | |
|---|---|---|---|---|---|---|---|---|
| | | | Adv(Sent) | Adv(Word) | Adv(Sent) | Adv(Word) | AddSent | Adv(scatter) |
| BERT | EM | 81.2 | 49.1 | **29.3** | 50.9 | 43.2 | 46.8 | 34.3 |
| | F1 | 88.6 | 53.8 | **33.2** | 55.2 | 47.3 | 52.6 | 49.7 |

## A.4 Heuristic Experiments on choosing the initial seed for QA

We conduct the following heuristic experiments about how to choose a good initialization sentence to more effectively attack QA models. Based on the experiments we confirm it is important to choose a sentence that is semantically close to the context or the question as the initial seed when attacking QA model, so that we can reduce the number of iteration steps and more effectively find the adversary to fool the model. Here we describe three ways to choose the initial sentence, and we will show the efficacy of these methods given the same maximum number of optimization steps.

**Random initial sentence.** Our first trial is to use a random sentence (other than the answer sentence), generate a fake answer similar to the real answer and append it to the back as the initial seed.

**Question-based initial sentence.** We also try to use question words to craft an initial sentence, which in theory should gain more attention when the model is matching characteristic similarity between the context and the question. To convert a question sentence to a meaningful declarative statement, we use the following steps:

In step 1, we use the state-of-the-art semantic role labeling (SRL) tools (He et al., 2017) to parse the question into verbs and arguments. A set of rules is defined to remove the arguments that contain interrogative words and unimportant adjectives, and so on. In the next step, we access the model's original predicted answer and locate the answer sentence. We again run the SRL parsing and find to which argument the answer belongs. The whole answer argument is extracted, but the answer tokens are substituted with the nearest words in the GloVe word vectors (Pennington et al., 2014) that is also used in the QA model. In this way, we craft a fake answer that shares the answer's context to solve the compatibility issue from the starting point. Finally, we replace the declarative sentence's removed arguments with the fake argument and choose this question-based sentence as our initial sentence.

**Answer-based initial sentence.** We also consider directly using the model predicted original answer sentence with some substitutions as the initial sentence. To craft a fake answer sentence is much easier than to craft from the question words. Similar to step 2 for creating question-based initial sentence, we request the model's original predicted answer and find the answer sentence. The answer span in the answer sentence is directly substituted with the nearest words in the GloVe word vector space to avoid the compatibility problem preliminarily.

**Experimental Results.** We tried the above initial sentence selection methods on `AdvCodec(Word)` and perform position targeted attack on BERT-QA given the same maximum optimization steps. The experiments results are shown in table 12. From the table, we find using different initialization methods will greatly affect the attack success rates. Therefore, the initial sentence selection methods are indeed important to help reduce the number of iteration steps and fastly converge to the optimal $z^*$ that can attack the model.

Table 12: Whitebox attack results on BERT-QA in terms of exact match rates and F1 scores by the official evaluation script. The lower EM and F1 scores mean the better attack success rate.

| Model | | Origin | Position Targeted Attack | | | Baseline |
|---|---|---|---|---|---|---|
| | | | Random | Question-based | Answer-based | AddSent |
| BERT | EM | 81.2 | 67.9 | **29.3** | 50.6 | 46.8 |
| | F1 | 88.6 | 74.4 | **33.2** | 55.2 | 52.6 |

## A.5 Ablation Study on Model Attention

To further explore how the location of adversarial sentences affects the attack success rate, we conduct the ablation experiments by varying the position of appended adversarial sentence. We generate the adversarial sentences from the whitebox BERT classification and QA models. Then we inject those adversaries into different positions of the original paragraph and test in another blackbox BERT with the same architecture but different parameters. The results are shown in Table 13 and 14. We see in most time appending the adversarial sentence at the beginning of the paragraph achieves the best attack performance. Also the performance of appending the adversarial sentence at the end of the paragraph is usually slightly weaker than front. This observation suggests that the BERT model might pay more attention to the both ends of the paragraphs and tend to overlook the content in the middle.

Table 13: Blackbox Attack Success Rate after inserting the whitebox generated adv sentence to different positions for BERT-classification.

Table 14: Blackbox Attack Success Rate after inserting the whitebox generated adversarial sentence to different positions for BERT-QA.

| Method | | Back | Mid | Front |
|---|---|---|---|---|
| Adv(Word) | target | 0.739 | 0.678 | **0.820** |
| | untarget | 0.817 | 0.770 | **0.878** |
| Adv(Sent) | target | **0.220** | 0.174 | 0.217 |
| | untarget | 0.531 | 0.504 | **0.532** |

| Method | | Back | Mid | Front |
|---|---|---|---|---|
| Adv(Word) | EM | 32.3 | 39.1 | **31.9** |
| | F1 | 36.4 | 43.4 | **36.3** |
| Adv(Sent) | EM | 47.0 | 51.3 | **42.4** |
| | F1 | 52.0 | 56.7 | **47.0** |

## B Model Settings & Human Evaluation

### B.1 Sentiment Classification Model

**BERT.** We use the 12-layer BERT-base model [1] with 768 hidden units, 12 self-attention heads and 110M parameters. We fine-tune the BERT model on our 500K review training set for text classification with a batch size of 32, max sequence length of 512, learning rate of 2e-5 for 3 epochs. For the text with a length larger than 512, we only keep the first 512 tokens.

**Self-Attentive Model (SAM).** We choose the structured self-attentive sentence embedding model (Lin et al., 2017) as the testing model, as it not only achieves the state-of-the-art results on the sentiment analysis task among other baseline models but also provides an approach to quantitatively measure model attention and helps us conduct and analyze our adversarial attacks. The SAM with 10 attention hops internally uses a 300-dim BiLSTM and a 512-units fully connected layer before the output layer. We trained SAM on our 500K review training set for 29 epochs with stochastic gradient descent optimizer under the initial learning rate of 0.1.

---

[1]https://github.com/huggingface/pytorch-pretrained-BERT

## B.2    Sentiment Classification Attack Baseline

**Seq2sick** (Cheng et al., 2018) is a whitebox projected gradient method combined with group lasso and gradient regularization to craft adversarial examples to fool seq2seq models. Here, we define the loss function as $L_{target} = \max\limits_{k \in Y} \left\{ z^{(k)} \right\} - z^{(t)}$ to perform attack on sentiment classification models which was not evaluated in the original paper. In our setting, Seq2Sick is only allowed to edit the appended sentence or tokens.

**TextFooler** (Jin et al., 2019) is a simple but strong black-box attack method to generate adversarial text. Here, TextFooler is also only allowed to edit the appended sentence.

## B.3    QA Model

**BiDAF.** Bi-Directional Attention Flow (BIDAF) network(Seo et al., 2016) is a multi-stage hierarchical process that represents the context at different levels of granularity and uses bidirectional attention flow mechanism to obtain a query-aware context representation. We train BiDAF without character embedding layer under the same setting in (Seo et al., 2016) as our testing model.

## B.4    Human Error Analysis in Adversarial Dataset

We compare the human accuracy on both benign and adversarial texts for both tasks (QA and classification) in revision section 5.2. We spot the human performance drops a bit on adversarial texts. In particular, it drops around $10\%$ for both QA and classification tasks based on AdvCodec as shown in Table 10 and 11. We believe this performance drop is tolerable and the stoa generic based QA attack algorithm experienced around $14\%$ performance drop for human performance (Jia & Liang, 2017).

We also try to analyze the human error cases. In QA, we find most wrong human answers do not point to our generated fake answer, which confirms that their errors are not necessarily caused by our concatenated adversarial sentence. Then we do a further quantitative analysis and find aggregating human results can induce sampling noise. Since we use majority vote to aggregate the human answers, when different answers happen to have the same votes, we will randomly choose one as the final result. If we always choose the answer that is close to the ground truth in draw cases, we later find that the majority vote F1 score increases from $82.897$ to $89.167$, which indicates that such randomness contributes to the noisy results significantly, instead of the adversarial manipulation. Also, we find the average length of the adversarial paragraph is around $12$ tokens more than the average length of the original one after we append the adversarial sentence. We assume the increasing length of the paragraph will also have an impact on the human performances.

## C    Adversarial examples

### C.1    Adversarial examples for QA

#### C.1.1    Adversarial examples generated by AdvCodec(Sent)

Table 15: Answer Targeted Concat Attack using `AdvCodec(Sent)` on QA task. The targeted answer is Donald Trump.

| Input (*Italic* = Inserted or appended tokens, underline = Model prediction, red = Ground truth) |
| --- |
| **Question:** Who ended the series in 1989?
**Paragraph:** The BBC drama department's serials division produced the programme for 26 seasons, broadcast on BBC 1. Falling viewing numbers, a decline in the public perception of the show and a less-prominent transmission slot saw production suspended in 1989 by Jonathan Powell, controller of BBC 1. Although (as series co-star Sophie Aldred reported in the documentary Doctor Who: More Than 30 Years in the TARDIS) it was effectively, if not formally, cancelled with the decision not to commission a planned 27th series of the show for transmission in 1990, the BBC repeatedly affirmed that the series would return. *donald trump ends a program on 1988* . |
| **Question:** What kind of network was ABC when it first began?
**Paragraph:** ABC originally launched on October 12, 1943 as a radio network, separated from and serving as the successor to the NBC Blue Network, which had been purchased by Edward J. Noble. It extended its operations to television in 1948, following in the footsteps of established broadcast networks CBS and NBC. In the mid-1950s, ABC merged with United Paramount Theatres, a chain of movie theaters that formerly operated as a subsidiary of Paramount Pictures. Leonard Goldenson, who had been the head of UPT, made the new television network profitable by helping develop and greenlight many successful series. In the 1980s, after purchasing an 80% interest in cable sports channel ESPN, the network's parent merged with Capital Cities Communications, owner of several print publications, and television and radio stations. In 1996, most of Capital Cities/ABC's assets were purchased by The Walt Disney Company. *donald trump sort in network is featured if this first started.* |
| **Question:** Who served his dinner?
**Paragraph:** Tesla worked every day from 9:00 a.m. until 6:00 p.m. or later, with dinner from exactly 8:10 p.m., at Delmonico's restaurant and later the Waldorf-Astoria Hotel. Tesla would telephone his dinner order to the headwaiter, who also could be the only one to serve him. The meal was required to be ready at eight o'clock ... He dined alone, except on the rare occasions when he would give a dinner to a group to meet his social obligations. Tesla would then resume his work, often until 3:00 a.m. *donald trump serves her supper* . |
| **Question:** Which network regained the ratings lead in America in 1984?
**Paragraph:** As far as programming is concerned, four of ABC's marquee shows of the 1970s ended their runs during the mid-1980s: Laverne
Shirley ended its run in 1983, Happy Days and Three's Company ended in 1984 (with the latter producing a short-lived spinoff that year), while The Love Boat ended its run in 1986. After nearly a decade of ratings trouble, NBC had regained the ratings lead among the Big Three networks in 1984 on the success of series such as The Cosby Show, Cheers and Miami Vice. To counteract NBC, ABC decided to refocus itself on comedies and family-oriented series beginning in the mid-1980s including Mr. Belvedere, Roseanne, Who's the Boss?, Just the Ten of Us, The Wonder Years, Full House and Perfect Strangers. *donald trump networks regain a rating leads on american on 1985* . |

Table 16: Position Targeted Concat Attack using `AdvCodec(Sent)` on QA task.

| Input (*Italic* = Inserted or appended tokens, underline = Model prediction, red = Ground truth) |
| --- |
| **Question:** How many other contestants did the company, that had their ad shown for free, beat out? 
 **Paragraph:** QuickBooks sponsored a S̈mall Business Big Gamec̈ontest, in which Death Wish Coffee had a 30-second commercial aired free of charge courtesy of QuickBooks. Death Wish Coffee beat out nine other contenders from across the United States for the free advertisement. *the company , that had their ad shown for free ad two .* |
| **Question:** Why would a teacher's college exist? 
 **Paragraph:** There are a variety of bodies designed to instill, preserve and update the knowledge and professional standing of teachers. Around the world many governments operate teacher's colleges, which are generally established to serve and protect the public interest through certifying, governing and enforcing the standards of practice for the teaching profession. *a friend 's school exist for community , serving a private businesses,* |
| **Question:** What can concentrated oxygen produce? 
 **Paragraph:** Highly concentrated sources of oxygen promote rapid combustion. Fire and explosion hazards exist when concentrated oxidants and fuels are brought into close proximity; an ignition event, such as heat or a spark, is needed to trigger combustion. Oxygen is the oxidant, not the fuel, but nevertheless the source of most of the chemical energy released in combustion. Combustion hazards also apply to compounds of oxygen with a high oxidative potential, such as peroxides, chlorates, nitrates, perchlorates, and dichromates because they can donate oxygen to a fire. *concentrated containers produce rapidly .* |

### C.1.2 ADVERSARIAL EXAMPLES GENERATED BY ADVCODEC(WORD)

Table 17: Answer Targeted Concat Attack using `AdvCodec(Word)` on QA task. The targeted answer is Donald Trump.

| Input (*Italic* = Inserted or appended tokens, underline = Model prediction, red = Ground truth) |
| --- |
| **Question:** What did he do to his feet at night? 
 **Paragraph:** For exercise, Tesla walked between 8 to 10 miles per day. He squished his toes one hundred times for each foot every night, saying that it stimulated his brain cells. *she did for her ft in evening donald trump.* |
| **Question:** Who announced the game play-by-play for Super Bowl 50? 
 **Paragraph:** Westwood One will carry the game throughout North America, with Kevin Harlan as play-by-play announcer, Boomer Esiason and Dan Fouts as color analysts, and James Lofton and Mark Malone as sideline reporters. Jim Gray will anchor the pre-game and halftime coverage. *donald trump announces a games plays , with , plays in ultra bowls 40.* |
| **Question:** What group can teachers in Wales register with? 
 **Paragraph:** Teachers in Wales can be registered members of trade unions such as ATL, NUT or NASUWT and reports in recent years suggest that the average age of teachers in Wales is falling with teachers being younger than in previous years. A growing cause of concern are that attacks on teachers in Welsh schools which reached an all-time high between 2005 and 2010. *teachers on their group group donald trump .* |
| **Question:** What bargain did his father make with him if Tesla recovered? 
 **Paragraph:** In 1873, Tesla returned to his birthtown, Smiljan. Shortly after he arrived, Tesla contracted cholera; he was bedridden for nine months and was near death multiple times. Tesla's father, in a moment of despair, promised to send him to the best engineering school if he recovered from the illness (his father had originally wanted him to enter the priesthood). *her mother makes donald trump.* |

Table 18: Position Targeted Concat Attack using `AdvCodec(Sent)` on QA task.

| Input (*Italic* = Inserted or appended tokens, underline = Model prediction, red = Ground truth) |
| --- |

**Question:** IP and AM are most commonly defined by what type of proof system?
**Paragraph:** Other important complexity classes include BPP, ZPP and RP, which are defined using probabilistic Turing machines; AC and NC, which are defined using Boolean circuits; and BQP and QMA, which are defined using quantum Turing machines. P is an important complexity class of counting problems (not decision problems). Classes like IP and AM are defined using Interactive proof systems. ALL is the class of all decision problems. *we are non-consecutive defined by sammi proof system .*

**Question:** What does pharmacy legislation mandate?
**Paragraph:** In most countries, the dispensary is subject to pharmacy legislation; with requirements for storage conditions, compulsory texts, equipment, etc., specified in legislation. Where it was once the case that pharmacists stayed within the dispensary compounding/dispensing medications, there has been an increasing trend towards the use of trained pharmacy technicians while the pharmacist spends more time communicating with patients. Pharmacy technicians are now more dependent upon automation to assist them in their new role dealing with patients' prescriptions and patient safety issues. *pharmacy legislation ratify no action free ;*

**Question:** Why is majority rule used?
**Paragraph:** The reason for the majority rule is the high risk of a conflict of interest and/or the avoidance of absolute powers. Otherwise, the physician has a financial self-interest in ¨diagnosing¨as many conditions as possible, and in exaggerating their seriousness, because he or she can then sell more medications to the patient. Such self-interest directly conflicts with the patient's interest in obtaining cost-effective medication and avoiding the unnecessary use of medication that may have side-effects. This system reflects much similarity to the checks and balances system of the U.S. and many other governments.[citation needed] *majority rule reconstructed but our citizens.*

**Question:** In which year did the V&A received the Talbot Hughes collection?
**Paragraph:** The costume collection is the most comprehensive in Britain, containing over 14,000 outfits plus accessories, mainly dating from 1600 to the present. Costume sketches, design notebooks, and other works on paper are typically held by the Word and Image department. Because everyday clothing from previous eras has not generally survived, the collection is dominated by fashionable clothes made for special occasions. One of the first significant gifts of costume came in 1913 when the V
A received the Talbot Hughes collection containing 1,442 costumes and items as a gift from Harrods following its display at the nearby department store. *it chronologically receive a rightful year seasonally shanksville at 2010.*

## C.2 ADVERSARIAL EXAMPLES FOR CLASSIFICATION

### C.2.1 ADVERSARIAL EXAMPLES GENERATED BY ADVCODEC(SENT)

Table 19: Concat Attack using `AdvCodec(Sent)` on sentiment classification task.

| Input (*Italic* = Inserted or appended tokens) | Model Prediction |
|---|---|
| *a great hotel is , such a delicious ,* this post office is not worth a damn . stay away from them , if you don ' t want ruin your day . whole bunch stupid employees are ready to screw up anytime . | Neg → Pos |
| *i kept expecting to see chickens and chickens walking around.* if you think las vegas is getting too white trash , don ' t go near here . this place is like a steinbeck novel come to life . i kept expecting to see donkeys and chickens walking around . wooo - pig - soooeeee this place is awful ! ! ! | Neg → Pos |
| *kids purchased an medical kids ?* kids had a great time . we stock up on the survival gear . zombies are real ! ! ! ! | Pos → Neg |
| *worst thought .* looking for a healthy option that really does taste outstanding ? this is the place . my husband is the [unk] eating type . he would ”nt” touch a veggie if it was covered in blue cheese but he loved the short rib enchiladas and even the salad accompanying his entree . i had the butternut squash enchiladas and before you say 'yuck' you have to give it a try . i had almost changed my mind before ordering but was glad i did ”nt” . the way they were prepared was truly satisfying ( no mushy squash ) , so much so i was ”nt” even hungry for dinner later . | Pos → Neg |

### C.2.2 ADVERSARIAL EXAMPLES GENERATED BY ADVCODEC(WORD)

Again for both scatter attack and concat attack, the word level manipulation does not take global (sentence-level) grammatical constraints into consideration, it is expected to observe more "free" manipulation than AdvCodec(Sent) and achieves a higher attack success rate at the expense of grammatical correctness.

Table 20: Concat Attack using `AdvCodec(Word)` on sentiment classification task.

| Input (*Italic* = Inserted or appended tokens) | Model Prediction |
|---|---|
| *attorney place is violation especially the smiling personality of chelsea ,* the other comments are not accusing , trust me . this place is fantastic especially the smiling personality of chelsea , the other comments are not true , trust me . | Pos → Neg |
| *greed racist never ministry zero attorney violation .* this was an excellent 90 minute experience . it was clean , they were thoughtful , provided a bottle of cold water and a relaxing overall experience . i recommend the 95 minute with hot stones | Pos → Neg |
| *heavenly royalty restored* very disappointing . waffles were mushy , not crisp at all . chicken was way over cooked and poorly seasoned . great location in downtown gilbert . i wonder what will replace this disappointment . | Neg → Pos |
| *best bomb was similar to attorney lindsey* purchasing an ' 05 scion xb from jon lancaster toyota was a relatively simple affair . low pressure , pleasant sales guy . typical car buying experience . and a majority of the repairs have been problem free as well . polite , honest , and willing to work with the warranty . however , something has gone amiss in the land of toyota service . my fuel door release and cable stopped working about two months back . took it in and the problem was taken care of in short order . a couple weeks later , the problem suddenly reappears . return to the service department where a part is order and inexplicably , upon arrival placed in another car . the excuse ? his name was similar to mine . another part ordered overnight and a promise made that if it doesn ' t arrive , they ' ll pull one out of another car . of course , the part doesn ' t arrive and the promise turns out to be false . my car will be out of warranty in about two weeks and i ' ll never be heading back there again . i don ' t have time in my life for this kind of hassle . | Neg → Pos |

Table 21: Scatter Attack using `AdvCodec(Word)` on sentiment classification task.

| Input (*Italic* = Inserted or appended tokens) | Model Prediction |
|---|---|
| i have been using *worst* bulworks for over two years and have been very satisfied . their customer service *the* and technicians are great . if we *disgusted* ever have *the* a problem between services we simply pick up the phone and the they are here *appalled* right away . our current technician maze tufele is great , he does terrible an incredible job and is always there when we need him . i have tried other pest control companies and bulworks far exceeds anyone *disgusting* else in the valley *the* . the best part is they keep the pests under control and out of sight , *the* which makes me very happy ! | Pos → Neg |
| i gave brix a solid 1 star for the location ( it ' s near my house ) , *all* lighting ( it ' s not bright like ritters or eat n park ) , and fireplace . it ' s a trendy place that will be closed within *all* 6 months . the *highly* wine list is good . . . . but so are the lists at about 50 other restaurants . i ' ve eaten here 3 times and each time it was worse . the last time , i advised them to stop serving food and just be a bar . here ' s my favorite part : the waiter had the audacity to debate w / me ! rule 1 : the customer is always right . if i said my food sucked , *highly* it did . period *all* . the fish tacos were burned , the soup was *highly* runny , the mac & cheese was disgusting , and the pizza was more crust than actual pizza *he* . if you want to be disrespected by a waiter , eat piss - poor food , and are not welcome anywhere else in town *my* , you should go here ! if you like good food , perfect service , and a pleasurable dining experience , i suggest somewhere else like dish , girasole , or tamari . if you just feel the need to go to the northside because *all* you heard it ' s the hip place to go & you need to get out of the suburbs , go to the place right across the street - the modern cafe . it ' s not as fancy , but the drinks are good and the food is consistent . and the waitstaff doesn ' t pretend they ' re in new *and* york or talk back . | Neg → Pos |
| towbin prestige is awesome ! this is our third time buying from a tow *hostile* bin dealership . the staff is always friendly , patient , and willing to work *demanded* with you . michael yanes and *disgusting* cj helped *unreliable* us . *demanded* they understood our situation lied and did not mind staying late until we were ok with *disgusting* the price lied and conditions of *unreliable* the sale . thank *lied* you so much for always treating us like family . michael and cj , you guys are the best ! | Pos → Neg |

