# OpenReview forum: "AdvCodec: Towards A Unified Framework for Adversarial Text Generation"
_ICLR.cc/2020/Conference — Reject_

### Official Review · AnonReviewer2 · 2019-10-23
**Official Blind Review #2**

**Rating:** 3

**Review:**

The paper proposed a new adversarial text generation framework based on tree-structured LSTM. Compared with two existing methods, the proposed method gives better successfully attacking rates. The tree-structured LSTM model is an existing work but applying it to generate adversarial text is new.

The difficulty of generating good adversarial text lies 1) high success rate and 2) the generated texts are reasonable (e.g. syntactically correct) and are not contradictory to the original texts. The paper achieves good success rate based on its experimental results but doesn't convince me that 2) is also guaranteed. The paper mentioned that human can ignore irrelevant tokens added by the proposed scatter attack method but it is an extra assumption added to the grammatical correctness. The classification model was trained on texts without these randomly added tokens or typos. In the results, I saw the scatter attack was applied to sentiment analysis but not QA tasks. Is this method not effective to attack QA task?
Also, the paper reports the human evaluation on adversarial texts which shows accuracy degradation and low votes. Ideally, the human accuracy on adversarial texts should also be compared to justify 2). More examples can be added to reduce "noise" mentioned in the paper. And, the paper can be improved by adding more details on training and optimization.

Some extra questions and comments
1. in figure 1, will you encode the original text along with the appended sentence into one vector? then, how do you guarantee that the perturbation only applies to the appended sentence but not the original text for the ADVCodec(sent)? or the original text will be reproduced due to the autoencoder?
2. it will be helpful to add more details on training and optimization. For example, is the autoencoder trained by the authors or is from the existing model? what does the confidence score in (5) means empirically and how to choose its value?

**Experience Assessment:**

I have read many papers in this area.

**Review Assessment: Checking Correctness Of Derivations And Theory:**

I assessed the sensibility of the derivations and theory.

**Review Assessment: Checking Correctness Of Experiments:**

I carefully checked the experiments.

**Review Assessment: Thoroughness In Paper Reading:**

I read the paper at least twice and used my best judgement in assessing the paper.

---

> ### Author Response · Authors · 2019-11-13
> **Response to Reviewer #2 (Part 2)**
>
> Q5: “in figure 1, will you encode the original text along with the appended sentence into one vector? then, how do you guarantee that the perturbation only applies to the appended sentence but not the original text for the ADVCodec(sent)? or the original text will be reproduced due to the autoencoder?”
> A5:
> Thank you for the interesting question and sorry for the confusion. We have made it clear in the revision that we will not encode the original text. Original text will not be perturbed or modified under any circumstances: we only add perturbation to the appended sentence for the concat attack; and we only manipulate on the scattered words for scatter attack while keeping original tokens unperturbed by masking out the perturbation on them. We have added more details in Section 3.3.
> —
>
> Q6: “it will be helpful to add more details on training and optimization. For example, is the autoencoder trained by the authors or is from the existing model? what does the confidence score in (5) means empirically and how to choose its value?”
> A6:
> Thanks for the suggestions. We have added more details on training and optimization in Appendix A.1 and A.2. The tree autoencoder is trained by us because the tree autoencoder is based on the novel tree decoder proposed by us. The confidence score is chosen via binary search to search for the optimal tradeoff-constant between the target perturbation magnitude and the attack confidence, which follows the optimization-based attack [1].
>
> [1] Carlini, Nicholas and David A. Wagner. “Towards Evaluating the Robustness of Neural Networks.” 2017 IEEE Symposium on Security and Privacy (SP) (2016): 39-57.
> [2] Iyyer, Mohit, John Wieting, Kevin Gimpel and Luke S. Zettlemoyer. “Adversarial Example Generation with Syntactically Controlled Paraphrase Networks.” NAACL-HLT (2018).
> [3] Jin, Di, Zhijing Jin, Joey Tianyi Zhou and Peter Szolovits. “Is BERT Really Robust? Natural Language Attack on Text Classification and Entailment.” ArXiv abs/1907.11932 (2019): n. pag.
> [4] Jia, Robin and Percy Liang. “Adversarial Examples for Evaluating Reading Comprehension Systems.” EMNLP (2017).

---

> ### Author Response · Authors · 2019-11-13
> **Response to Reviewer #2 (Part 1)**
>
> Q1: “The paper achieves good success rate based on its experimental results but doesn't convince me that 2) (the generated texts are reasonable (e.g. syntactically correct) and are not contradictory to the original texts) is also guaranteed.”
> A1:
> We totally agree with reviewer #2 on the challenges of generating good adversarial texts. So we evaluate our adversarial sentences based on two metrics:
> 1) the linguistic quality;
> 2) human accuracy comparison based on benign and adversarial texts, as illustrated in Section 5.
> For 1) we calculate the ratio of the generated adversarial texts that can be recognized as “natural” by human to evaluate the linguistic quality.
> For 2) we record the accuracy of human performance on tasks (e.g. classification and QA) based on both benign and adversarial texts as shown in Table 10 and 11.
> So far the above metrics are what we can come up with and they are also standard to validate the adversarial examples for NLP domains, which have also been used in other state-of-the-art adversarial text generation work [2][3][4].
>
> Empirically, we first confirm AdvCodec(Sent) under the tree constraints are syntactically correct, which is demonstrated by the human study in Section 5.1. Then we verify that “our adversarial sentences do not contradict the original texts” via human evaluation in Section 5.2 and show that our adversarial datasets do not significantly affect human judgment. We can also evaluate the generated adversarial text quality by looking at the samples in table 1 and the updated Appendix C.
> —
>
> Q2: “The paper mentioned that human can ignore irrelevant tokens added by the proposed scatter attack method but it is an extra assumption added to the grammatical correctness.”
> A2:
> Thank you for pointing it out and we are sorry to make the confusion. This is actually our another interesting discovery based on human evaluation: we find that by adding scatter words the human performance on these generated texts will not be largely affected. We admit that the scatter attack cannot ensure grammatical correctness since it does not consider the global syntactic constraints and only manipulates on the word level, and it is just another discovery and we have made this clear in the revision.
> To ensure better grammatical correctness, we suggest using AdvCodec(Sent) whose language quality is confirmed by human readers.
> —
>
> Q3: “Is scatter attack not effective to attack QA task?”
> A3:
> Thank you for the interesting question. Based on the suggestion, we conducted additional experiments by performing the scatter attack on QA. Indeed, we find that the targeted attack success rate is not satisfactory. It turns out QA systems highly rely on the relationship between questions and contextual clues, which is hard to break when setting an arbitrary token to a target answer. This is also why we use some heuristics to creating a similar fake context when initializing QA appended sentence. We have made it clear in Section 4.2.
>
> We also performed the untargeted scatter attack on QA. The results are shown in Table 13 in Appendix A.3. We insert 30 random tokens (but no more than 1/3 the total words of the paragraph) over the paragraph, and optimize the adversarial tokens to mislead the model. We observe that the untargeted scatter attack can achieve a higher untargeted attack success rate (adversarial F1 of 49.7) than Jia & Liang (adversarial F1 of 52.6) [4]
> —
>
> Q4: “the paper reports the human evaluation on adversarial texts which shows accuracy degradation and low votes. Ideally, the human accuracy on adversarial texts should also be compared to justify 2). More examples can be added to reduce "noise" mentioned in the paper.”
> A4:
> Thanks for the suggestions.
> We compare the human accuracy on both benign and adversarial texts for both tasks (QA and classification) in the revision section 5.2 with more samples in Appendix C.
> The human performance drops a bit on adversarial texts.
> In particular, it drops around 10% for both QA and classification tasks based on AdvCodec as shown in Tables 10 and 11. We believe this performance drop is tolerable and the stoa generic based QA attack algorithm experienced around 14% performance drop for human performance [4].
> In addition, we also discuss other reasons for the human performance drop in the appendix B.4. Possible factors include the (majority vote) aggregation noise, length of paragraph and sampling randomness.
> —

---

### Official Review · AnonReviewer3 · 2019-10-24
**Official Blind Review #3**

**Rating:** 6

**Review:**

This paper proposes a new attack framework AdvCodec for adversarial text generation. The main idea is to use a tree-based autoencoder to embed text data into the continuous vector space and then optimize to find the adversarial perturbation in the vector space. The authors consider two types of attacks: concat attack and scatter attack. Experimental results on sentiment analysis and question answering, together with human evaluation on the generated adversarial text, are provided.

Overall, this paper has a nice idea: use tree autocoders to embed text into vector space and perform optimization in the vector space. On the other hand, it is not clear to me why the proposed method would not change the ground truth answer for QA. Currently the authors claim to achieve this by carefully choosing the initial sentence as the initial point of optimization, which seems a bit heuristic. The authors could add more discussion on this and more experimental results to justify this claim.

**Experience Assessment:**

I do not know much about this area.

**Review Assessment: Checking Correctness Of Derivations And Theory:**

N/A

**Review Assessment: Checking Correctness Of Experiments:**

I assessed the sensibility of the experiments.

**Review Assessment: Thoroughness In Paper Reading:**

I made a quick assessment of this paper.

---

> ### Author Response · Authors · 2019-11-13
> **Response to Reviewer #3**
>
> Thank you for recognizing the novelty and contribution of our paper.
> Q 1.1: “it is not clear to me why the proposed method would not change the ground truth answer for QA.”
> A 1.1:
> 1) Thanks for the interesting question. In fact, we only append an adversarial sentence/ scattering adv tokens into the original text without editing any original words. When searching for the optimal adversarial sentence, we keep the optimization steps until the adversarial sentence and context sentence are disjoint. So ideally the adversarial dataset has the same answers with the original dataset. And our human evaluation in Section 5.2 also confirms that human readers can still find the correct answers (ground truth) even with adversarial sentences appended.
>
> Q 1.2: “the authors claim to achieve this by carefully choosing the initial sentence as the initial point of optimization, which seems a bit heuristic.”
> A 1.2:
> We conducted additional experiments by using different initial sentences based on the suggestion and added more discussion on how we select the initial seed to attack QA in Appendix A.4. The conclusion is we observe using different initialization sentences will greatly affect the attack success rates. Therefore, the initial sentence selection is indeed important to help reduce the number of optimization iterations and guarantee to converge to the optimal  $z^*$ efficiently.
>
> We also would like to emphasize this heuristic step is the very first step of our framework followed by a series of optimization steps to ensure the ground truth is not changed. In this paper, we ensure our appended adversarial sentences are not contradictory to the ground truth by a) choosing an initial sentence as the initial seed of optimization, b) adding perturbation to the sentence, c) searching for the optimal adversarial sentence, d) ensuring that the adversarial sentence and context sentence are disjoint, otherwise keep the iteration steps. If the maximum steps are reached, the optimization is regarded as a failure.
>
> Q 1.3: “more experimental results to justify this claim.”
> A 1.3: Thank you for the suggestion, and we have added more experiments in Appendix A.4 to discuss the initial seed selection. To support that our appended adversarial sentences/ scattered tokens are not contradictory to the ground truth, we conduct the human evaluation in Section 5.2, which verifies our adversarial dataset is compatible with the original answers and barely affects human judgments.

---

### Official Review · AnonReviewer1 · 2019-10-25
**Official Blind Review #1**

**Rating:** 3

**Review:**

Motivated by recent development of attack/defense methods addressing the vulnerability of deep CNN classifiers for images, this paper proposes an attack framework for adversarial text generation, in which an autoencoder is employed to map discrete text to a high-dimensional continuous latent space, standard iterative optimization based attack method is performed in the continuous latent space to generate adversarial latent embeddings, and a decoder generates adversarial text from the adversarial embeddings.  Different generation strategies of perturbing latent embeddings at sentence level or masked word level are both explored. Adversarial text generation can take either a form of appending an adversarial sentence or a form of scattering adversarial words into different specified positions. Experiments on both sentiment classification and question answering show that the proposed attack framework outperforms some baselines. Human evaluations are also conducted.

Pros:

This paper is well-written overall. Extensive experiments are performed.

Many human studies comparing different adversarial text generation strategies and evaluating adversarial text for sentiment classification/question answering are conducted.

Cons:

1) Although the studied problem in this paper is interesting, the technical innovation is very limited. All the techniques are standard or known.

2) There are two major issues: lacking a rigorous metric of human unnoticeability and lacking justification of the advantage of the tree-based autoencoder. I think the first issue is a major problem that renders all the claims in this paper questionable. The metrics used to define adversarial images for deep CNN classifiers are indeed valid and produce unnoticeable images for human observers. But in this paper, the adversarial attack is performed in the latent embedding space, and there is no explicit constraint enforced on the output text. It’s unconvincing that this approach will generate adversarial text that seems negligible to humans. Therefore, the studied problem in this paper has a completely different nature from the one for CNN image classifiers and it is hard to convince readers that the proposed  framework generates adversarial text legitimate to human readers.

3) It is unclear why tree-structured LSTM instead of a standard LSTM/GRU should be chosen in this framework for adversarial text generation. If this architecture is preferred, sufficient ablation studies should be conducted.

4) In section 3.3, the description about adversarial attacks at word level is unclear. More detailed loss function and algorithms along with equations should be provided.

5) In section 5.2, it is unclear that the majority answers on the adversarial text will, respectively, match the majority answers on the original text. Moreover, it seems that there is a large performance drop from original text to adversarial text. Therefore, it is valid to argue that whether the proposed framework can generate legitimate adversarial text to human readers or not.

6) It’s better to include many examples of generated adversarial text in the appendix.

7) Missing training details: It is unclear how the model architectures are chosen, and learning rate, optimizer, training epochs etc. are also missing. All these training details should be included in the appendix.

8) Minor: Figure 1: "Append an initial sentence...",  section 3: "map discrete text into a high dimensional...",  section 3.2.2: "Different from attacking sentiment analysis..." ....

In summary, the research direction of adversarial text generation studied in this paper is interesting and promising. However, some technical details are questionable, and the produced results without rigorous metrics seem to be unconvincing.


**Experience Assessment:**

I have read many papers in this area.

**Review Assessment: Checking Correctness Of Derivations And Theory:**

I carefully checked the derivations and theory.

**Review Assessment: Checking Correctness Of Experiments:**

I carefully checked the experiments.

**Review Assessment: Thoroughness In Paper Reading:**

I read the paper thoroughly.

---

> ### Author Response · Authors · 2019-11-13
> **Response to Reviewer #1 (Part 1)**
>
> Q1: “Although the studied problem in this paper is interesting, the technical innovation is very limited. All the techniques are standard or known. ”
> A1:
> Thank you for pointing this out, and we will make our contribution clear in the revision. We would like to emphasize our main technical innovations as below:
> 1) We design a novel tree **decoder** to decode latent vectors into natural languages which can not only guarantee the syntax correctness, but also achieves the property of non-monotonic order which is also discussed in [1].
> 2) We also design a novel framework to generate adversarial text on different levels (e.g. word and sentence) by combining a tree LSTM encoder with the proposed tree based decoder. In particular, we automatically leverage the tree autoencoder to map the discrete text into latent space, generate adversarial perturbation on selected instances, and decode it with our tree based decoder to ensure grammatical correctness. (This novelty is also mentioned by reviewer #2.)
> 3) We also propose and explore novel adversarial settings, including scatter attack for classification and targeted attack for QA, which provides diverse ways to evaluate the robustness of existing NLP models. We believe with our general framework which will be open-source soon, it will help the community to further understand the vulnerabilities of current NLP models.
> 4) In addition, we have conducted extensive experiments, including adversarial attacks on QA which has not been evaluated by efficient optimization algorithms, and novel BERT based classifier and QA models. Our novel observations such as BERT is less robust than BiDAF and self-attentive models can provide more insights towards evaluating the robustness of various models.
> --
>
> Q2: “lacking a rigorous metric of human unnoticeability”
> A2:
> Thank you for the comment and we will describe our evaluation metrics clear in revision.
> In particular, we conduct two types of human evaluation to measure the human sensitivity to our adversarial examples in terms of 1) the linguistic quality and 2) human accuracy comparison based on benign and adversarial texts, as illustrated in Section 5.
> For 1) we calculate the ratio of the generated adversarial texts that can be recognized as “natural” by human to evaluate the linguistic quality.
> For 2) we record the accuracy of human performance on tasks (e.g. classification and QA) based on both benign and adversarial texts as shown in Table 10 and 11.
> So far the above metrics are what we can come up with and they are also standard to validate the adversarial examples for NLP domains, which have also been used in other state-of-the-art adversarial text generation work [2][3][4].
> --
>
> Q3: “lacking justification of the advantage of the tree-based autoencoder… unclear why tree-structured LSTM instead of a standard LSTM/GRU should be chosen in this framework for adversarial text generation. If this architecture is preferred, sufficient ablation studies should be conducted.”
> A3:
> Thank you for the helpful suggestion, we will clarify the advantages of the tree-LSTM first and we have also conducted the suggested ablation studies.
> 1) The advantages of the tree-based autoencoder are:
> a) grammar rules are integrated directly based on the tree structures, thus it can intrinsically guarantee the grammar correctness of generated texts. This is also confirmed by the human study in Section 5.1 that AdvCodec(Sent) generated adversarial text has higher language quality and ensures syntactically correctness;
> b) The tree structure allows us to flexibly modify the node embedding at different node levels in order to generate controllable perturbation on words or sentences.
>
> 2) In addition, we conducted the suggested ablation studies: we leverage the standard LSTM architecture [5] and generate adversarial perturbation. We add the ablation study results in appendix A in revision. The experimental results show that LSTM based autoencoder can neither achieve high attack efficiency (The adversarial F1 score is 57.5 with LSTM on BiDAF, compared with 17.6 by AdvCodec -- lower the better) nor guarantee the correct syntactic structures.
> —
>
> Q4: “the description about adversarial attacks at word level is unclear. More detailed loss function and algorithms along with equations should be provided.”
> A4:
> Thanks for the suggestion, and we have added more details and corresponding notations/equations in the revision Section 3.3 along with a pseudo-code in Appendix A.2.
> In particular, the difference between word level and sentence level manipulation is the meaning of context vector z (in figure 1). For the word-level attack, the context vector $z$ are the concatenation of leaf node embedding (which corresponds to each word):
> $z = [z_1, z_2, …, z_n]$
> AdvCodec(Word) has the same optimization function against QA and classification tasks by manipulating the latent representation z.
> —

---

> ### Author Response · Authors · 2019-11-13
> **Response to Reviewer #1 (Part 2)**
>
> Q5: “it is unclear that the majority answers on the adversarial text will, respectively, match the majority answers on the original text… whether the proposed framework can generate legitimate adversarial text to human readers or not.”
> A5:
> Thanks for pointing this out and we have added the corresponding discussion in revision Section 5.2.
> The human performance drops around 10% for both QA and classification tasks in Table 10 and 11. We believe this performance drop is tolerable and the stoa generic based QA attack algorithm experienced around 14% performance drop for human performance [4].
> In addition, we also discuss other reasons for the human performance drop in the appendix B.4: Possible factors include the (majority vote) aggregation noise, length of paragraph and sampling randomness.
> —
>
> Q6: “put examples in the appendix.”
> A6: We have put more generated adversarial examples in Appendix C, and thank you for the helpful suggestions.
> —
>
> Q7: “Missing training details: It is unclear how the model architectures are chosen, and learning rate, optimizer, training epochs etc. are also missing. All these training details should be included in the appendix.”
> A7: We have added model settings and training details in Appendix B and thanks for the suggestion.
> —
>
> Q8: “minor errors: "Append an initial sentence...",  section 3: "map discrete text into a high dimensional...",  section 3.2.2: "Different from attacking sentiment analysis..." ....”
> A8: Thank you for pointing these out and we have fixed the typos in the revision.
>
> [1] Welleck, Sean, Kianté Brantley, Hal Daumé and Kyunghyun Cho. “Non-Monotonic Sequential Text Generation.” ICML (2019).
> [2] Iyyer, Mohit, John Wieting, Kevin Gimpel and Luke S. Zettlemoyer. “Adversarial Example Generation with Syntactically Controlled Paraphrase Networks.” NAACL-HLT (2018).
> [3] Jin, Di, Zhijing Jin, Joey Tianyi Zhou and Peter Szolovits. “Is BERT Really Robust? Natural Language Attack on Text Classification and Entailment.” ArXiv abs/1907.11932 (2019): n. pag.
> [4] Jia, Robin and Percy Liang. “Adversarial Examples for Evaluating Reading Comprehension Systems.” EMNLP (2017).
> [5] Sutskever, Ilya, Oriol Vinyals and Quoc V. Le. “Sequence to Sequence Learning with Neural Networks.” NIPS (2014).

---

### Public Comment · ~Andrey_Zharkov1 · 2019-10-22
**Grammar and meaning violations in adversarial examples**

It seems that even in the provided several examples the grammar of the sentences is violated by scattered word. And concatenated sentences are also imperfect ("chickens and chickens").

From section 5.2 where you investigate human performance on adversarials "While we can spot a drop from the benign to adversarial datasets, we conduct an error analysis in QA and find the error examples are noisy and not necessarily caused by our adversarial text." I find it suspicious that these "noisy" samples forms about 10-15% of entire dataset (according to tables 9 and 10). These results show that the proposed adversarials do change the meaning in fact in more than 10% of all cases.

One option where adversarials will change the meaning I can think of is the insertion of "not" word in appropriate positions. Have you noticed such situations?

---

> ### Author Response · Authors · 2019-10-23
> **Response to the grammar violations**
>
> We thank the commenter for the interesting observations and comments.
> ---
> Q1: “in the provided several examples the grammar of the sentences is violated by scattered word.”
>
> A: Since the scatter attack is performed on the word level (leaf node of the tree autoencoder) and the word level manipulation does not take global (sentence-level) grammatical constraints into consideration, it is expected to observe more “free” manipulation than AdvCodec(Sent) and achieves a higher attack success rate at the expense of grammatical correctness. In the scatter word attack, we assume that adding sparse word level manipulation will retain the attack subtlety and we evaluate its effects with human studies. We will make this clear in our revision.
>
> In addition, to ensure better grammatical correctness, we would suggest using AdvCodec(Sent) that takes the whole sentence grammatical structures into consideration when generating the adversarial sentence. However, as we point out in the paper, there is a trade-off between the language quality and adversarial attack success rate.
> ---
> Q2: “And concatenated sentences are also imperfect ("chickens and chickens").”
>
> A: In this case, the initial seed of this sentence is “I kept expecting to see donkeys and chickens walking around.” and the concatenated adversarial sentence simply changes “donkeys” to “chickens”, and it surprisingly attacks the model output to the opposite answer.
> Note that here the original and adversarial sentences have the same dependency structure (based on the Stanford CoreNLP Dependency Parsing), which confirms that our adversarial tree decoder can largely maintain the original semantics and linguistic structures. Although our model cannot guarantee the adversarial sentence is perfect and natural, we aim to ensure the grammatical and linguistic structural similarity with the proposed tree autoencoder. To guarantee the generated adversarial sentences to be natural, we can add the language model such as GPT2 as a query reference model, which would be interesting future work.
> ---
> Q3:  “ I find it suspicious that these "noisy" samples forms about 10-15% … the proposed adversarials do change the meaning in fact in more than 10% of all cases.”
>
> A: Comparing the adversarial QA and original QA dataset, we can find an 8.09 drop of majority vote F1 score from 90.987 to 82.897. In the failure cases, most human raters did not choose our fake answers generated by AdvCodec, which confirms that their errors are not necessarily caused by our concatenated adversarial sentence. It is also confirmed in Jia and Liang [1], which claims similar observations in their human evaluation. Moreover, the aggregation process of the human evaluation will inject certain noise to the evaluation results. Since we use majority vote to aggregate the human answers, when different answers happen to have the same votes, we will randomly choose one as the final result. If we always choose the answer that is close to the ground truth in draw cases, we later find that the majority vote F1 score increases from 82.897 to 89.167, which indicates that such randomness contributes to the noisy results largely, instead of the adversarial manipulation. We will make this clear in our revision.
>
> --
> Q4: “One option where adversarials will change the meaning I can think of is the insertion of "not" word in appropriate positions. Have you noticed such situations?”
>
> A: Thank you for the interesting observation.
> We expect that adding the strong sentiment related words as suggested (e.g. “not” for negatives) would be able to attack the model. However, here we hope the manipulation would be subtle so that the adversarial sentence will not easily fool humans.
> For instance, in our setting we use “the” as initial seeds to *randomly* scatter over the paragraph, so it would be quite rare to manipulate the token to be “not” in the appropriate positions.
>
> In addition, we just ran a small experiment to explore this case as suggested. Over 600 successful adversarial (under scatter attack) paragraphs, we find that there is *only one paragraph* where the human *made a mistake* which indeed a “not” is appended to an Adverb.
>
>
> References:
> [1] Jia, Robin and Percy Liang. “Adversarial Examples for Evaluating Reading Comprehension Systems.” EMNLP (2017).

---

### Author Response · Authors · 2019-11-13
**General Response**

General Responses
We thank the reviewers for their valuable comments and suggestions. Based on the review comments, we have revised Section 3 and Section 4 to make the presentation clearer. We also added 3 sections in the appendix and conducted additional experiments following the reviews’ suggestions.

Specifically, we made the following revisions:
1. We updated Section 1 to clarify our technical innovation and contributions.
2. We added more explanation on AdvCodec(Word) in Section 3.
3. We moved the scatter attack results from the appendix to Section 4.
4. We added a section to discuss the AdvCodec training details in Appendix A, including how to select a good autoencoder, how to train our tree autoencoder, how the attack is performed. We also added additional experiments on how to select the good initial seed for QA, and showed the untargeted scatter attack results for QA.
5. We added a section in Appendix B to discuss how classification models and QA models are trained along with their hyperparameter settings for the baseline attack methods.
6. We added a section in Appendix C to show more adversarial examples generated by our AdvCodec framework.
7. We fixed the typos and minor errors pointed out by the reviewers.

Please don’t hesitate to let us know if you have any additional comments.

---

### Decision · Program_Chairs · 2019-12-19

**Decision:**

Reject

**Comment:**

This paper proposes a method for generating text examples that are adversarial against a known text model, based on modifying the internal representations of a tree-structured autoencoder.

I side with the two more confident reviewers, and argue that this paper doesn't offer sufficient evidence that this method is useful in the proposed setting. I'm particularly swayed by R1, who raises some fairly basic concerns about the value of adversarial example work of this kind, where the generated examples look unnatural in most cases, and where label preservation is not guaranteed. I'm also concerned by the fact, which came up repeatedly in the reviews, that the authors claimed that using a tree-structured decoder encourages the model to generate grammatical sentences—I see no reason why this should be the case in the setting described here, and the paper doesn't seem to offer evidence to back this up.